# Extended-Spectrum ß-Lactamase-Producing *Escherichia coli* in Conventional and Organic Pig Fattening Farms

**DOI:** 10.3390/microorganisms10030603

**Published:** 2022-03-11

**Authors:** Katharina Meissner, Carola Sauter-Louis, Stefan E. Heiden, Katharina Schaufler, Herbert Tomaso, Franz J. Conraths, Timo Homeier-Bachmann

**Affiliations:** 1Friedrich-Loeffler-Institut, Federal Research Institute for Animal Health, Institute of Epidemiology, 17493 Greifswald-Insel Riems, Germany; kathimeissner@web.de (K.M.); carola.sauter-louis@fli.de (C.S.-L.); franz.conraths@fli.de (F.J.C.); 2Institute of Pharmacy, University of Greifswald, 17489 Greifswald, Germany; stefan.heiden@uni-greifswald.de (S.E.H.); katharina.schaufler@uni-greifswald.de (K.S.); 3Institute of Infection Medicine, Christian-Albrecht University and University Medical Center Schleswig-Holstein, 24118 Kiel, Germany; 4Friedrich-Loeffler-Institut, Federal Research Institute for Animal Health, Institute of Bacterial Infections and Zoonoses, 07743 Jena, Germany; herbert.tomaso@fli.de

**Keywords:** antimicrobial resistance, extended-spectrum ß-lactamase-producing *Escherichia coli*, pig, organic farming

## Abstract

Antimicrobial resistance is an increasing global problem and complicates successful treatments of bacterial infections in animals and humans. We conducted a longitudinal study in Mecklenburg-Western Pomerania to compare the occurrence of ESBL-producing *Escherichia* (*E.) coli* in three conventional and four organic pig farms. ESBL-positive *E. coli*, especially of the CTX-M type, were found in all fattening farms, confirming that antimicrobial resistance is widespread in pig fattening and affects both conventional and organic farms. The percentage of ESBL-positive pens was significantly higher on conventional (55.2%) than on organic farms (44.8%) with similar proportions of ESBL-positive pens on conventional farms (54.3–61.9%) and a wide variation (7.7–84.2%) on organic farms. Metadata suggest that the farms of origin, from which weaner pigs were purchased, had a major influence on the occurrence of ESBL-producing *E. coli* in the fattening farms. Resistance screening showed that the proportion of pens with multidrug-resistant *E. coli* was similar on conventional (28.6%) and organic (31.5%) farms. The study shows that ESBL-positive *E. coli* play a major role in pig production and that urgent action is needed to prevent their spread.

## 1. Introduction

Antimicrobials are produced by naturally occurring bacteria and fungi found in the environment [1]. Many clinically relevant antibiotic resistance genes have most likely evolved from genes of environmental bacteria [2]. A recently published study postulates that wildlife represents a previously unrecognized medium through which environmental antimicrobial resistance genes can be transferred to human and animal clinical pathogens [2]. A retrospective study showed that the occurrence of antibiotic resistance in clinical isolates from before the introduction of antibiotics was very rare [3]. However, antimicrobial resistance (AMR) is a natural phenomenon [2,4], with misuse and overuse of antibiotics representing one of the main factors that select for the emergence of AMR [5]. The AMR problem is not limited to human medicine, since part of the resistance burden in humans is influenced by the use of antimicrobials in livestock [6,7,8,9]. Due to the increasing development of bacterial resistance to important antimicrobial agents in recent decades, AMR is an issue of utmost importance in both human and veterinary medicine worldwide as it can lead to treatment failure against common infectious diseases [10].

In the European Union (EU), the same antibiotics may be used in organic farming as well as in conventional farming, but under different circumstances [11]. According to EU regulations, routine prophylactic use of antibiotics in organic animal production is not allowed (Regulation (EC) No 834/2007). Since 2006, antibiotics were no longer permitted as performance enhancers (Regulation (EC) 1831/2003). However, antibiotic usage to treat infection and avoid animal suffering is allowed but with longer withdrawal periods than in conventional production (Regulation (EC) No 834/2007). If the lifespan of an animal in organic farming is less than one year, the products of this animal may only be marketed as “organic” when it has been treated with an antibiotic no more often than once (Regulation (EC) No 889/2008). 

The production of extended-spectrum ß-lactamases (ESBL) and AmpC ß-lactamases by bacteria, especially in members of the enterobacteria family, is an important resistance mechanism. ESBL-producing bacteria were first isolated in the late 1970s [12] and detected for the first time in human clinical isolates a few years later [13]. The occurrence of ESBL/AmpC-producing *E. coli* in livestock (cattle, poultry, sheep and pigs) has been known for several years [14]. 

ESBLs are mainly produced by enterobacteria, especially *Klebsiella pneumoniae* and *E. coli* [15]. ESBL-producing *E. coli* have been isolated from the feces of both sick and healthy animals [16,17,18]. The most abundant ESBL genes are *bla*_CTX-M-1_, *bla*_CMY-2_ and *bla*_SHV-12_, with SHV-12 predominating in food-producing animals in Europe [19].

The first detection of ESBL-producing *E. coli* in pigs was reported from China in 2002, where pigs had been sampled at the slaughterhouse and CTX-M-producing *E. coli* were detected in 2% of the animals [14]. García-Cobos et al. showed that the proportion of CTX-M-1 in *E. coli* predominated in domestic pigs in Germany [20]. The prevalence of ESBL-producing *E. coli* in pig farms varies widely across Europe. While prevalences ranged between 1 and 80% on farm level, 15–100% of the pigs were infected [21]. The lowest proportions of resistant *E. coli* were recorded in Norway, Sweden and Finland, whereas prevalences were higher in Spain, Italy, France, the Netherlands, Denmark and Belgium [22]. 

In a longitudinal study conducted in Denmark in 2011, it was shown that the proportion of CTX-M-producing *E. coli* decreased during the production cycle of the pig. Both bacteriological examination and polymerase chain reaction (PCR) showed that most pigs were CTX-M-positive shortly before weaning, while finishers were less affected [23].

A number of studies identified risk factors for the occurrence of AMR on farms. Several factors besides the use and overuse of antibiotics, which triggers the development of resistance, promote the spread of resistant bacteria and their genes locally and globally [24]. These include poor infection control, environmental contamination as well as the movement of infected people and animals [25].

Dohmen et al. showed that ESBL-*E. coli* were less frequent in pigs when drinking water was obtained from a public rather than a private source, when a hygiene sluice was the only access to the herd and when pest control was carried out by specialists [25]. Hering et al. identified the presence of a sick pen, underfloor ventilation in the stables and the use of insecticides against flies as risk factors [26]. In addition, the presence of wild birds in the vicinity of the farm, especially waterfowl, was associated with the presence of ESBL/AmpC-producing bacteria. Furthermore, pig farms in the vicinity of the farm were identified as a risk factor [27].

Most cross-sectional and longitudinal studies on ESBL/AmpC-producing *E. coli* in pigs conducted to date have taken place in conventional farms. Little is known about the prevalence of ESBL/AmpC-producing *E. coli* in organic farms. Fertner et al. demonstrated that purchase of weaner piglets from only one supplier is associated with a low use of antibiotics in fattening farms [28]. This fact, and also the legal restrictions on antibiotic use in organic pig farming, led to our hypothesis that the proportion of antibiotic-resistant pathogens is lower on organic than on conventional pig farms. We, therefore, aimed at investigating the occurrence of ESBL/AmpC-producing *E. coli* on organic pig farms in comparison to conventional farms over one year. Furthermore, the occurring ESBL/AmpC-carrying *E. coli* were characterized. The farms in Mecklenburg-Western Pomerania were sampled on several occasions and data on farm management, pig trade, the health status of the pigs and the use of antibiotics were collected and analyzed.

## 2. Materials and Methods

### 2.1. The Study

The study was conducted in Mecklenburg-Western Pomerania from February to December 2018. Suitable farms keeping a minimum of 500 and a maximum of 5000 fattening pigs were identified and contacted by official veterinarians in the respective districts. Farms participated voluntarily on the basis of informed consent of the farm manager. The AMR status of the farms was not known before the study was conducted. 

### 2.2. Pig Holdings

In total, three conventional and four organic farms participated in the study. One of the organic farms (farm 4) consisted of two locations which were 25 km apart and, therefore, were considered as two separate epidemiological units (farms 4a and 4b). As described in the introduction, EU regulations allow antibiotic usage in organic pig farming. Compared to conventional farms, however, there are some restrictions in the frequency of therapy and longer withdrawal periods. The locations of the farms are shown in Figure 1.

### 2.3. Sampling and Data Collection

Each farm was sampled on five occasions, i.e., approximately every two months. The aim was to sample six evenly distributed age groups on each farm. If possible, the groups were sampled on several occasions. If less than six age groups were present, all age groups were sampled.

Pooled fecal samples were taken from each sampled age group by filling 50 mL Falcon tubes with feces from each age group. If there were less than six age groups, ten 50 mL Falcon tubes were used per group. Sampling was evenly distributed over the whole stable or pen. In the organic farms, half of the samples were collected in the pens and the others in the runs, if possible. The samples were directly transported to the laboratory. The samples were cooled with cooling packs and kept in a cool box. 

Farm-related information was collected in structured interviews with the farm manager using a standard questionnaire (Appendix A). In particular, information on the antibiotics used and the frequency of applications is included.

During each farm visit, the farm manager was asked if diseases had occurred in the herd since the last sampling, if antibiotics had been used or other changes had been made (e.g., new suppliers of pigs or feed or changes in feeding the pigs).

### 2.4. Bacteriological Examination

Upon arrival at the laboratory, swabs (Sigma Transwab-Liquid Amies, Medical Wire & Equipment, Corsham, UK) were immediately taken from the fecal samples for bacteriological examination. The swabs were stored in the refrigerator at 6 °C until further processing, while the tubes were frozen at −20 °C.

The swab samples were streaked by dilution plating on selective agar plates, CHROMagar Orientation (Mast Diagnostica GmbH, Reinfeld, Germany), to which cefotaxime (2 µg/mL, Alfa Aesar, Thermo Fisher Scientific, Kandel, Germany) had been added. CHROM ID agar plates are particularly suitable for the identification of *E. coli* [29] and the supplementation with cefotaxime allows the identification of ESBL/AmpC-producing *E. coli* with high specificity [30]. We successfully applied this method in a recently published study [9]. The plates were incubated for 18–24 h at 37 °C. According to the manufacturer’s protocol, pink-violet colored, shiny colonies represent ESBL/AmpC-*E. coli*-positive results. One characteristic colony per swab was selected for further processing and sub-cultivated on CHROM ID agar plates supplemented with cefotaxime (2 µg/mL), until a pure culture was obtained. Due to the large number of isolates and the focus of the study (i.e., identifying differences between conventional and organic farming of fattening pigs and risk factors related to the burden of ESBL-*E. coli*), no further differentiation of isolates was performed.

An ESBL confirmation test (MASTDICS Combi Extended-Spectrum ß-Lactamases (ESßL) set, Mast Diagnostica GmbH, Reinfeld, Germany) was used to confirm the ESBL phenotype. The test was conducted according to the manufacturer’s specifications as well as the Clinical Laboratory Standards Institute (CLSI) guidelines [31]. Due to the high specificity for the detection of ESBL-producing *E. coli* of the bacteriological assay method [7,8,9], only a sample of 38 isolates was analyzed in the ESBL confirmatory test. 

For the resistance screening test, LB agar plates were spiked with clinical breakpoint concentrations of five different antimicrobial substances ([32]; ciprofloxacin was taken from CLSI M100 ED29:2019 [33]). These were ampicillin (≥32 µg/mL), tetracycline (≥16 µg/mL), streptomycin (≥64 µg/mL), gentamicin (≥16 µg/mL) and ciprofloxacin (≥1 µg/mL). The pure cultures from the bacteriological examination were spread on Columbia agar +5% sheep blood. Colony material was then collected using sterile toothpicks and transferred to the LB plates containing antibiotics. The plates were incubated at 37 °C for 24 h. If no bacterial growth was visible, the isolate was considered sensitive to the corresponding antibiotic. In contrast, the isolate was considered resistant if colony growth had occurred. This method was used as a screening test to identify isolates for further phenotypic resistance testing using a microdilution assay performed in the VITEK 2 Compact.

Animals close to slaughtering are of particular public health concern. In this age group, a total of 51 putative ESBL-producing *E. coli* were isolated from the fattening farms. These 51 isolates were further analyzed using VITEK2 Compact. To this end, the isolates were confirmed as *E. coli* by MALDI TOF-MS first [34]. The VITEK 2 Compact assay was conducted according to the manufacturer’s instructions. The assay was performed using the software version 9.02 and the AST-195 and AST-248 cards. 

For the interpretation of MIC values (sensitive, intermediate or resistant), clinical breakpoints according to CLSI M100 ED29:2019 [33] (ampicillin, amoxicillin/clavulanic acid, piperacillin, piperacillin/tazobactam, cefuroxime, cefotaxime, ceftazidime, cefepime, gentamicin, amikacin, tobramycin, ciprofloxacin, fosfomycin, nitrofurantoin, trimethoprim, trimethoprim/sulfamethoxazole, aztreonam, imipenem, meropenem, and ertapenem) or EUCAST 10.0 (Breakpoint Tables for the Interpretation of MICs and Zone Diameters (version 10.0, 2020. http://www.eucast.org, last accessed 9 October 2020) moxifloxacin, tigecycline, and colistin) were applied.

### 2.5. Data Analysis and Statistics

The results of the bacteriological examinations were carried out at pen-level, since pooled feces samples were taken from each pen. If a phenotypically cefotaxime-resistant *E. coli* strain was found in a pen, the pen, and thus, the animal group located there were defined as suspected ESBL- and AmpC-producing. The percentage of phenotypically cefotaxime-resistant pens was calculated for the conventional and organic farms as well as for the individual farms. To differentiate between age groups, quartiles were calculated. The fattening pigs were divided into four age groups of equal sizes based on quartiles. 

To analyze the potential association between the administration of antibiotics and results of bacteriological tests indicating the presence of ESBL- and AmpC-producing *E. coli*, the percentage of positive pens with and without antibiotic administration was calculated and compared for conventional and organic farms. Furthermore, possible differences regarding individual animal or group treatments were investigated. The results from the organic farms were analyzed for potential differences between ESBL-positive pens and runs. 

Comparisons were first made using the Fisher exact test, although we cannot rule out that pigs were sampled more than once; therefore, the results obtained on different sampling days may not have been independent. In a further step, a multivariable comparison with hierarchical structure was carried out. Using Microsoft Excel, 95% confidence intervals according to Clopper and Pearson were calculated [35]. 

For a quantitative between-farm comparison regarding potentially multidrug-resistant isolates, the percentages of ESBL-positive isolates against a maximum of five antibiotic classes were calculated in relation to the total of all samples. We defined an isolate as multidrug-resistant if it was resistant to at least three antibiotic classes. For each farm, the percentage of pens with multidrug-resistant bacteria was calculated in relation to all isolates used in the resistance screening test and the corresponding pens. Furthermore, the percentage ratios of pens with bacteria that were multidrug-resistant against all five antibiotic classes used in the resistance-screening test were determined for each farm in relation to all isolates used in the resistance-screening test and the corresponding pens. Since ciprofloxacin is of particular relevance due to its belonging to the group of “Highest Priority Critically Important Antimicrobials”, the percentages of ciprofloxacin-resistant pens in relation to all isolates used in the resistance-screening test were calculated for all farms.

Since some animal groups within the farms were sampled on several occasions, a dependency of the repeated samples could not be excluded. Therefore, we analyzed the potential effect of various influence variables on the result of the bacteriological examination (positive or negative) as the outcome variable in a generalized estimating equation (GEE) model using the package “gee” [36] in the statistical software R (R Core Team (2018). R: A language and environment for statistical computing. R Foundation for Statistical Computing, Vienna, Austria. Available online at https://www.R-project.org/, accessed on 1 December 2020).

In addition, GEE models were calculated to analyze potential associations between the age groups or the farm orientation (organic or conventional) and the occurrence of multidrug-resistant bacteria (resistance to at least three or five antibiotic groups). The significance level was set at *p* < 0.05.

## 3. Results

### 3.1. Questionnaire

There were large differences in the number of piglet suppliers among the individual farms. Some farms produced their own piglets (farms 2 and 7), while other farms had up to four different origins of the piglets. A major finding of the survey was that the health status of the piglets supplied varied widely. For example, piglets frequently suffered from respiratory diseases (farms 3 and 4b) or diarrhea (farms 5 and 6) at delivery. The results of the questionnaire are included in the Appendix A.

### 3.2. Bacteriological Examination

The percentage of ESBL-positive pens was greater on conventional farms, 55.2% (74 out of 134; 95% CI: 46.8–63.4), than on organic farms, 44.8% (60 out of 134; 95% CI: 36.6–53.2). The difference is not statistically significant (Fisher exact test, *p* = 0.112). A comparison of the individual farms showed that the range of values of the percentage of ESBL-positive pens was narrow (54.3–61.9%) on conventional farms (farms 1 and 3), while it varied between 7.7 and 84.2% on organic farms (farms 4a–7) (Figure 2). The proportion of positive pens showed statistically significant differences between the farms (Fisher exact test, *p* < 0.001). While the differences between the conventional farms were not statistically significant (Fisher exact test, *p* = 0.825), those between the organic farms were statistically significant (Fisher exact test, *p* < 0.001). Details are given in Table 1.

On farms 3, 4b and 6, group treatments with antibiotics were carried out in the sampled fattening groups in addition to individual animal treatments during the sampling period. In farm 3, 33.3% (4 of 12; 95% CI: 13.8–60.9) of the fattening groups, in farm 4b, 20% (2 of 10; 95% CI: 5.7–51.0) and in farm 6, 28.6% (2 of 7; 95% CI: 8.2–64.1) of the fattening groups were affected. The percentage of ESBL-positive fattening groups was greater on farms with group treatments (68.6%, 35 out of 51; 95% CI: 55.0–80.0) than on farms with individual treatments (49.7%; 95 out of 191; 95% CI: 42.7–56.8). The difference was statistically significant (Fisher exact test, *p* = 0.018). 

In addition, on the organic farms, the individual pens and runs were compared to each other. On farm 4b, the runs were only completed or ready for use from the third farm visit onwards and have thus not been sampled before. The percentage of ESBL-positive pens was 40.8% (49 out of 120; 95% CI: 32.5–49.8), while 34.3% (36 out of 105; 95% CI: 25.9–43.8) of the runs were ESBL-positive. The difference was not statistically significant (Fisher exact test, *p* = 0.337).

### 3.3. Resistance Screening

A total of 238 *E. coli* were used in the resistance screening test for multidrug resistance (i.e., resistant to penicillin, cephalosporins and at least one other class of antibiotics—MDR; [37]) and ciprofloxacin resistance. Except for three isolates that did not show resistance to any antibiotic, all isolates were at least resistant to penicillin and cephalosporins. The number of resistant isolates decreased as the number of antibiotic classes, to which resistance was present, increased (Table 2).

Multidrug-resistant isolates (i.e., resistant to penicillin, cephalosporins and at least one other class of antibiotics—3-MDR; [37]) were found on all farms except for farm 4a. The organic farms 4b and 6 showed the highest percentage of pens with 3-MDR bacteria. The conventional farm 3 also had almost 50% of pens with 3-MDR bacteria. In the pens of the other farms (both conventional and organic), the resistances were below 30%. The comparison of conventional and organic farms showed that the proportion of 3-MDR positive pens was almost the same. While it was 28.6% (34 out of 119; 95% CI: 21.2–37.3) in the conventional farms, 31.5% (35 out of 111; 95% CI: 23.6–40.7) positive pens were found in the organic farms. The difference was not statistically significant (Fisher exact test, *p* = 0.667).

Multidrug-resistant isolates against penicillin, cephalosporins and the other three classes of antibiotics used in the resistance screening test (5-MDR) were found in all farms except for farm 4a and 7. The comparison of the farms with regard to 5-MDR positive pens showed that the organic farms 4b and 6, as well as the conventional farm 3, had the most pens with 5-MDR bacteria. In all other farms, the proportion of pens with 5-MDR bacteria was below 10%. The comparison of conventional and organic farms showed that the proportion of pens with 5-MDR bacteria was 5.9% (7 out of 119; 95% CI: 2.9–11.7) on the conventional farms, while it was higher on the organic farms with 9.4% (10 out of 107; 95% CI: 5.2–16.4). The difference was not statistically significant (Fisher exact test, *p* = 0.450).

Ciprofloxacin belongs to the fluoroquinolones and is thus assigned to the “Highest Priority Critically Important Antimicrobials” [38]. The comparison of conventional and organic farms showed that the percentage of pens with *E. coli* resistant to ciprofloxacin was 9.2% (11 out of 119; 95% CI: 5.2–15.8) on conventional farms, while it was 17.3% (19 out of 110; 95% CI: 11.4–25.4) on organic farms. The difference was not statistically significant (Fisher exact test, *p* = 0.080).

### 3.4. VITEK 2 Compact AST Examination of Isolates Obtained from Pigs at the End of the Fattening Period

Animals in the finishing age group are of particular importance for public health, as these animals are awaiting slaughter. In this age group, a total of 51 putative ESBL-producing *E. coli* were isolated from the fattening farms. These 51 isolates were then analyzed for phenotypic resistance using VITEK2 Compact. AST results are shown in Table 3.

From farm 1, six isolates of pigs from the final fattening stage were investigated. Two of these isolates were resistant to penicillin and cephalosporins (including 3rd generation cephalosporins). However, two isolates were also resistant to 4th generation cephalosporins (cefepime), monobactams (aztreonam), diaminopyrimidines (trimethoprim and trimethoprim/sulfamethoxazole), aminoglycosides (gentamicin) and fluoroquinolones up to 4th generation (ciprofloxacin and moxifloxacin) and had an intermediate reaction to tobramycin (aminoglycoside). Overall, four different resistance profiles were identified in the six isolates.

Four isolates from finishing pigs were identified from farm 2. All showed different resistance profiles. The isolate with most resistances was resistant to penicillin, 4th generation cephalosporins, aminoglycosides (gentamicin), fluoroquinolones (ciprofloxacin and moxifloxacin), diaminopyrimidines (trimethoprim and trimethoprim/sulfamethoxazole) and monobactams (aztreonam). Furthermore, it was intermediate to tobramycin (aminoglycoside).

Six isolates were available from farm 3. Four isolates were resistant to penicillin and 3rd generation cephalosporins as well as to diaminopyrimidines (trimethoprim and trimethoprim/sulfamethoxazole). Table 3 shows the three different resistance profiles.

While there were no isolates from finishing pigs in farm 4a, 16 ESBL-carrying isolates were found in farm 4b. Three isolates were resistant to penicillin, cephalosporins (including 4th generation cephalosporins), fluoroquinolones (ciprofloxacin and moxifloxacin), aminoglycosides (gentamicin), diaminopyrimidines (trimethoprim and trimethoprim/sulfamethoxazole) and monobactams (aztreonam). Furthermore, they were intermediate to aminoglycosides (tobramycin). A total of eight different resistance profiles were identified.

On farm 5, 18 isolates were identified as ESBL. A total of 15 isolates showed an identical resistance profile. These isolates were resistant to penicillin and cephalosporins (including 3rd generation cephalosporins). One isolate was resistant to moxifloxacin (fluoroquinolone). Overall, three different resistance profiles were found, which are shown in Table 3.

From farm 6, 3 isolates were obtained from finishing pigs. The resistance profile of these isolates was almost identical. Overall, all isolates were resistant to penicillin, 3rd generation cephalosporins, fluoroquinolones (moxifloxacin) and monobactams (aztreonam). Two of these isolates were also resistant to cefepime (4th generation cephalosporin). There were two different resistance profiles.

On farm 7, no ESBL carrying *E. coli* isolates could be obtained in the finishing age group. 

In total, there were three farms (farms 1, 2 and 4b) where isolates from finishing pigs showed resistance to ciprofloxacin. Resistance to tobramycin was found in farm 2. Resistance to moxifloxacin was identified on five farms (farms 1, 2, 4b, 5 and 6). The comparison of all farms showed that farm 5 had the least resistance in terms of the number of antibiotics of the resistance profiles. *E. coli* from all farms were sensitive to amoxicillin/clavulanic acid, piperacillin/tazobactam, carbapenems, fosfomycin, nitrofurantoin, amikacin, tigecycline and colistin.

### 3.5. Multivariable Analysis

Multivariable analysis using the GEE model revealed a statistically significant difference between conventional and organic pig farms with regard to the results of the bacteriological examination (*p* < 0.001). Organic farms were less likely to be positive in the bacteriological testing than conventional farms. The age group comparison showed that age group 3 was statistically significantly different from age group 1 (*p* = 0.043), indicating a lower risk of age group 3 as being bacteriologically positive, while the other age groups were not statistically significantly different from age group 1.

A comparison of conventional and organic farms in relation to 3-MDR showed that there were tendencies for differences in the statistical evaluation for the organic farms (*p* = 0.059) and in age group 4 (*p* = 0.086), but these did not reach the level of statistical significance. The organic farms tended to have a higher odds ratio of being resistant to at least three classes of antibiotics than the conventional farms (OR = 2.136, 95% CI 0.971–4.698) and age group 4 tended to have a lower risk than age group 1.

When comparing conventional and organic farms, multivariable analysis revealed no statistically significant differences with regard to 5-MDR. 

When conventional and organic farms were compared with regard to resistance to ciprofloxacin, it became evident that the organic farms were statistically significantly more resistant to ciprofloxacin (*p* = 0.007) than the conventional farms. However, there were no statistically significant differences with regard to the age groups.

## 4. Discussion

### 4.1. Bacteriological Examination

At the farm level, the prevalence of ESBL-positive pig farms was 100% (8 of 8). Thus, it is higher than in other studies conducted in Germany. In 2012, Dahms et al. investigated five pig farms in Mecklenburg-Western Pomerania for their ESBL-status and identified three ESBL-positive farms (60%, 3 of 5) [39]. During the same period, Friese et al. also conducted investigations in pig farms in the north and east of Germany, where the prevalence was even lower (43.8%, 7 of 16) [16]. Furthermore, Hering et al. detected cefotaxime-resistant *E. coli* in 83% (40 of 48) of fecal samples from fattening pig farms investigated throughout Germany. However, it should be noted that the isolates were not confirmed as ESBL-producers and could have been positive for AmpC [26]. 

The differences between the work presented here and the results of the aforementioned studies might be due to the fact that the published data came from cross-sectional studies and, therefore, the pigs had only been sampled on a single occasion. As a result, it could be possible that due to fluctuations in resistance status, individual farms were negative at the (one-time) sampling date. Furthermore, these studies were conducted approximately six years ago. Thus, the results obtained in our study might even indicate an increase in prevalence, which may have been caused in particular by the ongoing frequent administration of antibiotics to fattening pigs. ESBL-producing pathogens will never disappear and they can be expected to continue to spread worldwide in the future, even in the One Health context [15]. 

Although all farms used antibiotics, a comparison of conventional and organic farms showed that the percentage of ESBL-positive pens was greater on conventional farms, 55.2% (74 of 134; 95% CI: 46.8–63.4), than on organic farms, 44.8% (60 of 134; 95% CI: 36.6–53.2). In addition, multivariable analysis using GEE modeling confirmed that there was a statistically significant difference between the two types of farms when comparing conventional and organic pig farms (*p* < 0.001). Conventional farms had a higher probability of being positive in bacteriological testing than organic farms. Intestinal *E. coli* resistance was less common in organic pig farms than in conventional farms in Denmark, France, Italy and Sweden as shown by Österberg et al. [40]. Studies in the USA [41] and New Zealand [42] came to similar conclusions. Dahms et al. studied four organic pig farms in Mecklenburg-Western Pomerania, Germany. All farms were ESBL-positive. However, there was no statistically significant effect on ESBL status, regardless of whether the farms were conventional or organic [39]. 

In addition, a comparison of individual farms illustrated that the percentage of ESBL-positive pens was very similar among conventional farms (54.3–61.9%), while it varied widely among farms managed organically, ranging from 7.7 to 84.2%. 

Dohmen et al. showed that the probability of finding ESBL-producing *E. coli* on a farm was greater when piglets were purchased. This result was not statistically significant, but a trend could be observed [25]. Fertner et al. showed that pig farms with a low use of antibiotics, among others, purchased weaner piglets from only one supplier [28]. The impact of the farms of origin on the ESBL status of the fattening pig farms became also evident in our study when the organic farms 4a and 4b were compared. Although the same person was responsible for the management of both farms, they showed the greatest difference of all farms overall in terms of the proportions of ESBL-positive pens. One major difference between the two farms was that they obtained their piglets from different suppliers. Thereby, diseases continuously occurred in the fattening pigs in farm 4b. The pigs regularly had diseases of the respiratory tract, they often had problems with joint inflammations, tail biting and cannibalism, as well as regularly occurring diarrhea. For this reason, not only individual animals received antibiotics in farm 4b, but they were sometimes used in the whole fattening group. This may have triggered an increase in ESBL-positive pens. On farm 4a, the fattening pigs also showed diseases of the respiratory tract, but they were milder and rarely required the use of antibiotics, which were only administered to individual animals. A similar situation was present on farm 7. The farm produced its own piglets and had the lowest proportion of ESBL-positive pens following farm 4a.

These results suggest that not only the fact that the pigs who were purchased from external suppliers may be associated with a higher proportion of ESBL-positive pens, but also that the situation regarding existing diseases on the farm of origin may play a decisive role. Hansen et al. showed that there was an association between the CTX-M status of a sow and that of her piglets (however, this was not statistically significant in the study) [23]. This illustrates how important animal health status and farm management, especially in terms of hygiene and biosecurity, already seems to be in producer farms.

Resistance screening showed that the proportion of pens with multidrug-resistant *E. coli* (resistant to at least three classes of antibiotics) was very similar between conventional (28.6%) and organic (31.5%) farms. The results of multivariable analysis indicated that organic farms tended to have a higher risk of being resistant to at least three classes of antibiotics (*p* = 0.059). This may be due to closer contact to the surroundings in the runs, which allow contact with soil, puddles and wild bird droppings. Further characterization of the corresponding isolates, e.g., by whole genome sequencing, might be helpful to determine whether there are breeding style-specific differences. This is planned for follow-up studies.

Similar results were obtained when conventional and organic farms were compared with respect to pens with multidrug-resistant pathogens (resistant to all five of the antibiotic classes used in the resistance screening test). Here, the percentage was slightly higher on organic (9.4%) than on conventional farms (5.9%). However, no statistically significant difference could be shown by multivariable analysis. 

Very few studies have investigated multidrug-resistance in pigs comparatively between conventional and organic farms. One group compared *E. coli* strains derived from pork and found that the incidence of MDR *E. coli* strains was significantly (*p* < 0.0001) higher in conventional pork than in organic pork [43]. In the study by Gebreyes et al., the occurrence of MDR *Salmonella* bacteria in antibiotic-free and conventional pig farms was one of the factors investigated. The authors found that multidrug-resistant strains were also present on farms that had never used any antibiotics to treat pigs. They hypothesized that several risk factors (e.g., usage of copper [44]) allow resistant strains to persist even without selection pressure from antibiotics [45]. 

We were unable to clarify conclusively why the proportions of pens with multidrug-resistant bacteria were higher in the organic farms than in the conventional pig fattening units we studied. However, it is necessary to differentiate between individual farms: the proportions of pens with multidrug-resistant germs (resistant to at least three as well as to all five antibiotics) were highest on only two organic farms (farm 4b and 6). 

Farm 4b had both the highest proportion of ESBL positive pens and also the highest proportions of pens with multidrug-resistant bacteria. In contrast, the proportions of pens with multidrug-resistant germs were proportionally smaller in organic farms 5 and 7 with respect to ESBL-positive pens. The conventional farm 3 had the highest proportion of pens with multidrug-resistant bacteria of the conventional farms. This might be due to the use of a large variety of different antibiotics. Six different antibiotics were used in weaners, while another four active substances were administered to fattening pigs. Another reason may be that not only individual animal treatments were carried out, but antibiotics were also administered to whole groups. It is known from the literature that weaners, followed by suckling piglets, received more antibiotics compared to fattening pigs. Thus, it was especially noticeable in suckling piglets that many 3rd and 4th generation cephalosporins had been given to them. Overall, piglets frequently received aminopenicillins, macrolides and polymyxins [46].

Multivariable analysis revealed another statistically significant difference between the two breeding types in addition to the aforementioned higher likelihood of being ESBL-positive in a conventional farm. When we looked at the percentage of barns with ciprofloxacin-resistant *E. coli*, the proportion of barns in conventional farms (9.2%) was lower than in organic farms (17.3%) (*p* = 0.007). However, the resistance of these strains to ciprofloxacin did not necessarily develop on the fattening farms. Particularly on organic farms, there are several possible entry routes for these pathogens through the runs (e.g., contact with soil, puddles and wild bird droppings). Recent studies demonstrated that ciprofloxacin-resistant *E. coli* are present in wild animals and that they are introduced into the environment, e.g., via wastewater from slaughterhouses [7,8]. For *mcr*-carrying *E. coli*, a meta-analysis revealed that environmental samples have the highest prevalence of these pathogens [47]. 

As already described in the section “bacteriological examination”, the two farms 4a and 4b are particularly interesting; both farms were managed by the same person, but obtain the piglets from different suppliers with considerable differences in piglet health. As for the proportions of ESBL-positive compartments, these two farms also harbored the maximum (farm 4b) and the minimum (farm 4a) of pens that were positive for multidrug-resistant and ciprofloxacin-resistant *E. coli*. Moreover, for both the proportions of multidrug-resistant and ciprofloxacin-resistant *E. coli*, the lowest proportions of positive compartments were found on farms with few external piglet suppliers (farm 1) or their own piglet production (farms 2 and 7).

The presence of multidrug resistance often accompanies fluoroquinolone resistance [48]. Often, the individual resistance genes are part of the same plasmids [49]. Whole genome sequencing could clarify whether this is also the case in the isolates of the present study. In addition, it could be screened for other genes of specific traits (e.g., biofilm formation) and examined to discover whether clones occur within a stable. 

Organic farmers were aware of the fact that they often purchased piglets that brought diseases from the farm of origin into the fattening sector. However, they pointed out that there were very few piglet producers in the organic sector and that sometimes there were not enough fattening piglets on offer, so they had no alternative other than to buy piglets that were at risk of bringing diseases and possibly multidrug-resistant bacteria with them.

### 4.2. VITEK 2 Compact Examination

Whether transmission of ESBL-producing *Enterobacteriaceae* to humans can occur via meat consumption is controversially discussed in the literature. For example, some studies have found similarities between human isolates and those from meat, suggesting that transmission is possible [50,51,52,53]. On the other hand, Irrgang et al. showed that further molecular epidemiological investigations revealed a great diversity of CTX-M-1 positive isolates [54]. 

The aim of the present study was to investigate which resistance profiles occur in the finishing age group and can potentially enter the food chain. Therefore, all ESBL-positive isolates from finishing pigs were analyzed in the study using VITEK 2 Compact. These animals were slaughtered soon after sampling and the examination of the isolates was aimed at identifying resistances that could potentially be transmitted to humans. 

All isolates were sensitive to carbapenems. Carbapenemase-producing bacteria currently cause major concern because they are often resistant to other antibiotics as well, and thus, leave only a few therapeutic options [55]. 

Although farms 1, 2, 4b and 7 did not use aminoglycosides in fattening pigs, isolates with resistance to gentamicin were found in these farms. In addition, bacteria resistant to tobramycin were detected on farm 2. This could be due to co-resistance or because the piglet producers that delivered animals to farms 1 and 4b had already administered aminoglycosides to the piglets. All isolates were sensitive to amikacin. 

Isolates resistant to the fluoroquinolones ciprofloxacin (farms 1, 2 and 4b) and moxifloxacin (farms 1, 2, 4b, 5 and 6) were found in many farms. Both active substances must not be used in food-producing animals. Farm 4b, which did not use fluoroquinolones in its fattening pigs, was particularly conspicuous. Nevertheless, 7 of 16 isolates showed resistance to moxifloxacin. 

All isolates were sensitive to fosfomycin and tigecycline. Both agents are classified as reserve antibiotics and are only used in human medicine. 

Although colistin was used in some farms, isolates from all farms were sensitive to this compound. A similar picture in terms of prevalence was found in German fattening pigs (1%) and fattening pigs in the finishing period (0%) [56]. However, in 2015, Liu et al. found plasmid-mediated colistin resistance genes (*mcr*-1) in *E. coli* isolates from animals, food and human sepsis samples from China [57]. Additionally, *mcr*-1 carrying ESBL-*E. coli* could be detected in wastewater from pig slaughterhouses [7].

## 5. Conclusions

The present study shows that antibiotic resistance is widespread in fattening pigs. ESBL-producing *E. coli* were present in both conventional and organic farms. The burden of ESBL-*E. coli* varied greatly from farm to farm. Farms with only one piglet supplier or their own piglet production tended to have lower loads of ESBL-positive, multidrug-resistant and ciprofloxacin-resistant *E. coli*. However, the proportion of multidrug-resistant and ciprofloxacin-resistant pathogens was not lower on organic farms than on conventional farms.

The focus of the study was to investigate differences between conventional and organic fattening pigs and to derive risk factors associated with the burden of ESBL-producing *E. coli*. For this reason, and because of the large number of isolates obtained, we focused the study on determining phenotypic resistance. To gain more detailed insights into individual groups (e.g., ciprofloxacin-resistant isolates from this study along with isolates from other farms), we plan to perform whole genome sequencing in the future. These investigations can then contribute to a better understanding of the sources of entry and intra-farm spread, and help to define intervention measures.

## Figures and Tables

**Figure 1 microorganisms-10-00603-f001:**
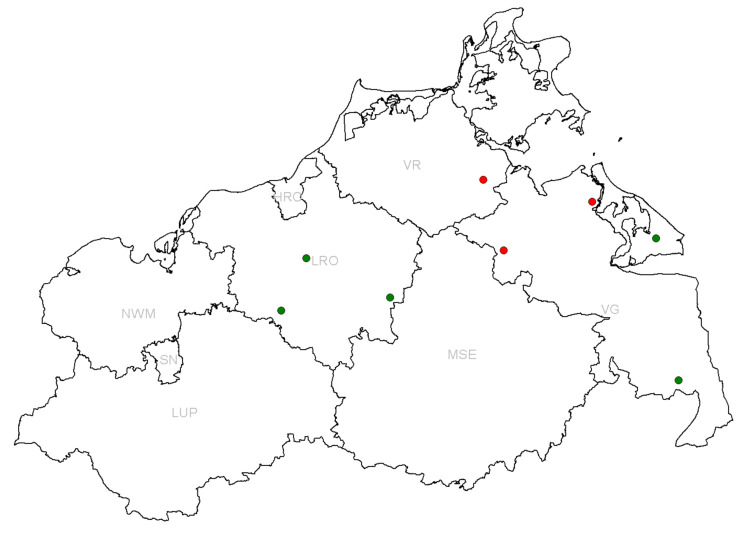
Location of sampled farms in the federal state of Mecklenburg-Western Pomerania. Conventional fattening farms are depicted in red and organic ones in green. The district names are indicated in light gray (HRO = Hansestadt Rostock, LRO = Rostock-Land, LUP = Ludwigslust-Parchim, MSE = Mecklenburgische Seenplatte, NWM = Nordwest-Mecklenburg, SN = Schwerin, VG = Vorpommern-Greifswald, VR = Vorpommern-Rügen).

**Figure 2 microorganisms-10-00603-f002:**
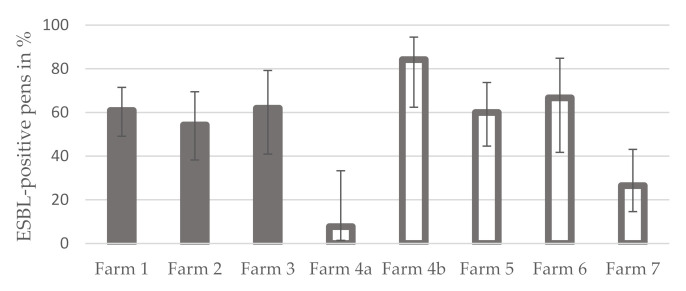
Average proportion of ESBL-positive pens per fattening pig farm in Mecklenburg-Western Pomerania (grey bars = conventional farms, grey bordered bars = organic farms; error indicators represent the 95% confidence intervals).

**Table 1 microorganisms-10-00603-t001:** Proportions of ESBL-positive pens and runs in organically managed fattening pig farms in Mecklenburg-Western Pomerania per farm and farm visit in percent.

	Sampling 1	Sampling 2	Sampling 3	Sampling 4	Sampling 5
Farm 1	100% (13/13)	100% (14/14)	6.7% (1/15)	28.6% (4/14)	76.9% (10/13)
Farm 2	100% (7/7)	28.6% (2/7)	71.4% (5/7)	0.0% (0/7)	71.4% (5/7)
Farm 3	66.7% (2/3)	100% (4/4)	16.7% (1/6)	100% (4/4)	50.0% (2/4)
Farm 4a Pen Run	0.0% (0/2) 0.0% (0/2)	50.0% (1/2) 0.0% (0/3)	0.0% (0/2) 0.0% (0/2)	0.0% (0/3) 0.0% (0/3)	0.0% (0/3) 0.0% (0/3)
Farm 4b Pen Run	100% (2/2) -	66.7% (2/3) -	75.0% (3/4) 100% (3/3)	100% (5/5) 100% (3/3)	60.0% (3/5) 33.3% (1/3)
Farm 5 Pen Run	50.0% (2/4) 25.0% (1/4)	71.4% (5/7) 0.0% (0/2)	57.1% (4/7) 50.0% (3/6)	9.1% (1/11) 18.2% (2/11)	81.8% (9/11) 63.6% (7/11)
Farm 6 Pen Run	100% (3/3) 100% (3/3)	75.0% (4/4) 100% (4/4)	0.0% (0/4) 25.0% (1/4)	33.3% (1/3) 0.0% (0/3)	100% (1/1) 100% (1/1)
Farm 7 Pen Run	0.0% (0/6) 33.3% (2/6)	16.7% (1/6) 33.3% (2/6)	28.6% (2/7) 42.9% (3/7)	0.0% (0/7) 0.0% (0/7)	12.5% (1/8) 0.0% (0/8)

**Table 2 microorganisms-10-00603-t002:** Bacteriologically positive samples in fattening pigs and results of resistance screening. 3-MDR = isolates resistant to penicillin and cephalosporins and at least one other class of antibiotics; 5-MDR = isolates resistant to penicillin and cephalosporins and at least three other classes of antibiotics.

Farm	Number of Tested Samples	Resistance Screening			
		Number of Tested Isolates *	3-MDR	5-MDR	Resistant to Ciprofloxacin
1	106	41	15 (36.6%)	2 (4.9%)	4 (9.8%)
2	98	34	11 (32.4%)	2 (5.9%)	3 (8.8%)
3	113	23	20 (87.0%)	4 (17.4%)	5 (21.7%)
4a	113	0	0 (0.0%)	0 (0.0%)	0 (0.0%)
4b	132	48	46 (95.8%)	7 (14.6%)	19 (39.6%)
5	240	58	13 (22.4%)	1 (1.7%)	2 (3.4%)
6	134	24	23 (95.8%)	2 (8.3%)	12 (50.0%)
7	200	10	9 (90.0%)	0 (0.0%)	7 (70.0%)

* i.e., phenotypically cefotaxime-resistant *E. coli*.

**Table 3 microorganisms-10-00603-t003:** Resistance profiles in the VITEK 2 Compact study of 51 isolates derived from pigs in finishing fattening (≥90 d in fattening). Isolates with identical results in AST from one farm were summarized to a resistance profile. (Con = conventional farm; Org = organic farm, S = sensitive, I = intermediate, R = resistant).

	Farm 1 (Con)				Farm 2 (Con)				Farm 3 (Con)			Farm 4b (Org)								Farm 5 (Org)			Farm 6 (Org)	
Ampicillin	R	R	R	R	R	R	R	R	R	R	R	R	R	R	R	R	R	R	R	R	R	R	R	R
Amoxicillin/Clavulanic acid	S	S	S	S	S	S	S	S	S	I	S	S	S	S	S	S	S	S	S	S	S	S	S	S
Piperacillin	R	R	R	R	R	R	R	R	R	R	R	R	R	R	R	R	R	R	R	R	R	R	R	R
Piperacillin/Tazobactam	S	S	S	S	S	S	S	S	S	S	S	S	S	S	S	S	S	S	S	S	S	S	S	S
Cefuroxime	R	R	R	R	R	R	R	R	R	R	R	R	R	R	R	R	R	R	R	R	R	R	R	R
Cefotaxime	R	R	R	R	R	R	R	R	R	R	R	R	R	R	R	R	R	R	R	R	R	R	R	R
Ceftazidime	R	S	S	S	R	S	S	S	S	S	S	R	S	R	S	R	S	S	S	S	R	S	S	S
Cefepime	R	S	S	S	R	S	S	S	S	S	S	R	S	S	S	S	S	S	S	S	S	S	R	S
Gentamicin	R	S	S	S	R	S	S	I	S	S	S	R	S	S	S	S	S	S	S	S	S	S	S	S
Amikacin	S	S	S	S	S	S	S	S	S	S	S	S	S	S	S	S	S	S	S	S	S	S	S	S
Tobramycin	I	S	S	S	I	S	S	R	S	S	S	I	S	S	S	S	S	S	S	S	S	S	S	S
Ciprofloxacin	R	S	S	S	R	S	S	S	S	S	S	R	R	S	S	S	S	S	S	S	S	S	S	S
Moxifloxacin	R	S	S	S	R	S	S	S	S	S	S	R	R	R	S	R	S	R	S	S	S	R	R	R
Fosfomycin	S	S	S	S	S	S	S	I	S	S	S	S	S	S	S	S	S	S	S	S	S	S	S	S
Nitrofurantoin	S	S	S	S	S	S	S	S	S	S	S	S	S	S	S	S	I	S	S	S	S	S	S	S
Trimethoprim	R	R	S	S	R	R	R	S	R	S	S	R	S	S	S	R	S	S	S	S	S	S	S	S
Trimethoprim/Sulfamethoxazole	R	R	S	S	R	R	R	R	R	S	S	R	S	S	S	R	S	S	S	S	S	S	S	S
Aztreonam	R	R	R	S	R	R	S	I	S	S	S	R	R	S	S	R	S	S	S	S	S	S	R	R
Imipenem	S	S	S	S	S	S	S	S	S	S	S	S	S	S	S	S	S	S	S	S	S	S	S	S
Meropenem	S	S	S	S	S	S	S	S	S	S	S	S	S	S	S	S	S	S	S	S	S	S	S	S
Ertapenem	S	S	S	S	S	S	S	S	S	S	S	S	S	S	S	S	S	S	S	S	S	S	S	S
Tigecycline	S	S	S	S	S	S	S	S	S	S	S	S	S	S	S	S	S	S	S	S	S	S	S	S
Colistin	S	S	S	S	S	S	S	S	S	S	S	S	S	S	S	S	S	S	S	S	S	S	S	S

## Data Availability

The data presented in this study are available in this article and the Appendix A.

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
