# Peer review of "Extended-Spectrum ß-Lactamase-Producing Escherichia coli in Conventional and Organic Pig Fattening Farms"

_microorganisms, 2022, doi:10.3390/microorganisms10030603_

Round 1

Reviewer 1 Report

This is a nice and high-quality work related to the hot topic of antibiotic-resistance of bacteria and the possibility of transmission to humans. The designed work-flow is ideal and the statistical analysis adequate.

I have a question why the authors could not ensure a unique sampling of piglets? Still, a random secondary sampling of a few piglets does not change statistical analysis significantly.

The study reveals how important is nowadays the correct tracking of the health status of incoming piglets into a farm and the certification of the health status of piglets in every producer. Thus, it would be easier to control diaspora and transmission of antibiotic-resistant bacterial strains.

Furthermore, in Line 563 you say "The assumed generally lower burden of ESBL-E. coli on organic 564 farms was confirmed in our study". I believe this statement is too strong to be included as I do not see that in Figure 2 and there were no statistically significant differences. In addition, Farm 4a can really affect the results as there is a significant difference in health status compared to the other organic farms.

Finally, please change all the subheadings (3.1...3.2..3.4..etc) to Bold fond in order to make it easier for the readers.

I suggest the acceptance of the paper after applying the few minor revisions.

Congrats for the work!

Reviewer 2 Report

Katharina Meissner (microorganisms-1624298), presented an interesting studying regarding the Extended-spectrum ß-lactamase-producing bacteria between two farming styles. However, there are many issues that should be clarified or better addressed before the publication.

Major:

the definition for the two farming styles should be clearly documented, and how it is was conducted in your study. I am always curious how much antibiotics were used, for numerous purposes, or no antibiotics at all for organic in this case.

Is there a way to further study the genotypic difference for the bacteria, I would suggest a WGS-based approach, this could possibly indicate the mechanism and ways for an explanation of the numerous questions in this study. 

What is the limitation for this study, and the future direction in the field should be clearly and well documented here in the discussion?

Minor:

Line 96-97. is there any association between herd size and AMR or CTX-M carrying bacteria.

Line 101. What is the exact difference for conventional and organic farms? Do they all use antibiotics for therapeutical purposes or growth-promoting purposes... How much antibiotics did they use or any types? Is there any other or specific feed additive used in these farms?

Line 139. "E. coli were identified on the basis of their 140 colony morphology" Are there other ways, i.e. molecular or Mass spectrometry approach, to confirm if it is E.coli. Morphology is not always working.

Line 147-148. what is the number difference between 38 and 238? and later 51. what is the relationship, are there some overlap among these numbers? It is very confusing for this study. I would suggest a mindmap for the study design how the bacteria were sampled and which bacteria were split for certain studies.

Line 149-158. this is not a typical antimicrobial resistance susceptibility assay.  I would suggest a broth dilution assay for this experiment. please be accurate!

Line 160-161. which drugs were tested, and Line 166. how about the result interpretation? there are two methods, CLSI and EUAST, which drug is followed by these approaches. Please be exact.

Line 235. are there seasonal issue on samplings, I saw some type of trend.

Line 267. certain level of repetition between table 2 and following figuures, fig 3-5.

LIne 463-465, an interesting observation that should be addressed, I am wondering if the bacteria differ between two types of breeding styles.

Line 498-504. the section should add more detailed information, which could make possible explanations, regarding the somewhat unexpected findings.

There is a better reference that could be used here (10.3390/microorganisms7100461), which describes a bigger picture of the problem.

Line 561-563. please turn down your tense for your conclusion here.

Line 583. some references are in the wrong style and if possible, should be replaced by new literature (10-20 years old??). I suggest the literature within 5 years in the field should take a major part.

Round 2

Reviewer 2 Report

none